# Time Series Models of the Human Heart in Patients with Heart Failure: Toward a Digital Twin Approach

**DOI:** 10.3390/s26010082

**Published:** 2025-12-22

**Authors:** Nilmini Wickramasinghe, Nalika Ulapane, Yuxin Zhang, Paul Jansons, Gunnar Cedersund, Ralph Maddison

**Affiliations:** 1School of Computing, Engineering & Mathematical Sciences, La Trobe University, Melbourne, VIC 3086, Australia; 2Faculty of Health, Deakin University, Melbourne, VIC 3125, Australia; yuxin.zhang@deakin.edu.au (Y.Z.); paul.jansons@deakin.edu.au (P.J.); ralph.maddison@deakin.edu.au (R.M.); 3Department of Biomedical Engineering (IMT), Linköping University, SE-581 83 Linköping, Sweden; gunnar.cedersund@liu.se

**Keywords:** artificial intelligence, chronic disease, digital twin, heart failure, machine learning, personalized care, precision medicine, regression, time series, wearable sensors

## Abstract

Digital Twins (DTs) are digital replicas of physical entities. The use of DTs in healthcare is a growing area of research. With DTs, there is potential to revolutionize healthcare with the assistance of Artificial Intelligence. This can lead to achieving precision, personalization, and value addition in healthcare. Contributing to this field, we present one of the first attempts of uncovering time series models of decompensation of heart failure. This was performed using some of the first data collected from the pilot phase of the SmartHeart study, in which an at-home, wearable, wireless sensor-based digital self-monitoring system for people with heart failure was tested.

## 1. Introduction

Digital twins (DTs) are typically described as digital replicas of entities in the physical world [1]. DTs are used to simulate the characteristics of a physical entity to assist with intelligent decision making, and this is enabled through data transfer and communication, ideally in real time. This concept has been revolutionizing many industries, including sectors such as aerospace, control engineering, smart cities, product design, and smart manufacturing to note a few. More recently, this concept has been explored in the healthcare sector as well, mainly with aims such as improving clinical decision making [2,3], improving the precision and personalization of care [4,5,6], and optimizing clinical workflows [7,8,9].

In parallel with advances of DTs, the Internet and the Internet of Things (IoT) have advanced over the years, and IoT devices and sensors are now being utilized to monitor people in the comfort of their home [10], delivering value-added and personalized healthcare. SmartHeart study [11] is a study carried out to evaluate an at-home, wearable, wireless sensor-based digital self-monitoring system for people with heart failure. In this paper, we present some of the first data collected from the SmartHeart study. Using this data, we present one of the first attempts of uncovering underlying time series models of decompensation of heart failure. We discuss some preliminary observations made from the estimated models. We thereby propose the possibility of using sensors at home to capture longitudinal data, estimate underlying models, and study their variations over time. Studying such variations over a sufficient period may benefit us through giving insights into the diversity of different people and even enable us to forecast various health events. Although we have not proceeded at this stage to the extent of analyzing this data over a long period of time and predicting cardiac events, in this paper, we present a computationally simple approach usable for uncovering underlying models of heart failure decompensation in real time. Uncovering such models can lead to the realization of DTs of people, especially the heart [12,13,14], over time, and these can serve as useful proxies for monitoring patients and tracking their progression, diversities, and similarities to eventually target precise and personalized care. While our work with heart failure is a case study, our approach can be generalized to longitudinal monitoring of various other health conditions.

## 2. Sensor Setup

The sensor setup used in SmartHeart [11] is depicted in Figure 1 and was designed to enable self-monitoring and thereby improve the safety and quality of life of people with heart failure. The various sensors used along with the parameters measured are summarized in Table 1.

## 3. Method

From the three main sources of measurements (i.e., heartrate, body weight, and blood pressure) collected from the dedicated sensor setup, we opted to model the heart using the heartrate, as that source of data is a form of continuous data collected using Smartwatches. Suppose the heartrate of a person (in beats per minute) is measured every minute. If the minute is indexed as $k,\;k\in Z^{+}$, suppose the heartrate measured in that minute is denoted as $h_{k},$ where $h_{k}\in Z^{+}$. Suppose we have a set of historical heartrate measurements such as $\left\{h_{1},\;h_{2},\;h_{3},\;\ldots ,\;h_{k}\right\}$ collected from a person. This set of measurements is essentially a time series. One of the simplest ways to model a time series is the autoregressive model architecture [15]; as such, we opted to the autoregressive architecture in Equation (1), in which we attempt to estimate the heartrate in the next minute, i.e., $h_{k+1}^{*}$, using a set of adjacent historical heartrates. In Equation (1), we set k≥2m+1 to ensure the estimation of reliable models and the predictions being reliable. The set $\left\{a_{0}^{\left(k\right)},\;a_{1}^{\left(k\right)},\;a_{2}^{\left(k\right)},\;\ldots a_{m}^{\left(k\right)}\right\}$ denotes the real-valued model parameters estimated from the previous k heartrates. Given that the chosen model architecture can be estimated efficiently, we can afford to estimate a new model each minute prior to predicting the heartrate for the next minute. This iterative model estimation also helps us assess model convergence to ensure reliability. We arrange the model parameters as in Equation (2), where *T* denotes the matrix transpose. We iteratively estimate the model parameters for each minute by solving the minimization problem in Equation (3).(1)$$h_{k+1}^{*}=a_{0}^{\left(k\right)}+\sum_{i=1}^{m}a_{i}^{\left(k\right)}h_{k+1-i}$$(2)$$a^{\left(k\right)}={\left[a_{0}^{\left(k\right)},\;a_{1}^{\left(k\right)},\;a_{2}^{\left(k\right)},\;\ldots \;a_{m}^{\left(k\right)}\right]}^{T}$$(3)$$a^{\left(k\right)}=\underset{a^{\left(k\right)}}{\mathrm{argmin}}\sum_{i=m+1}^{k}{\left(h_{i}^{*}-h_{i}\right)}^{2}$$

From the data collected from the SmartHeart study [11], we have so far encountered 23 people who had more than one day worth of data, i.e., more than 1440 min or 1440 points of heartrate observations. For our preliminary modeling, we collected the first 1440 heartrate observations from these participants. Thereby, we set the maximum value for k+1 to be 1440, i.e., k≤1439. For computational simplicity, we set the maximum for m to be 5. The value of m dictates how many adjacent historical heartrate values are considered in the model. For example, m=1 means the model considers only the heartrate of the previous minute to predict the heartrate of the next minute. Conversely, m=5 means the model considers the heartrate of the previous five minutes. The numbers in between can be interpreted accordingly. Within those constraints, we attempted to estimate 23 models that best describe the heartrate of the 23 corresponding people.

Since we estimate models iteratively, we computed the mean absolute difference between $h_{i}^{*}-h_{i}$ for the last 20% of the predictions, as shown in Equation (4), as a proxy for the model error. The model error is denoted as Err, and l denotes the smallest integer value that is greater than 80% of 1439, subject to the constraint k≤1439 we set earlier. Similarly, to assess the convergence of model parameters, we computed the mean and the standard deviation for each parameter in the most accurate model estimated for each of the 23 people.(4)$$Err=\frac{1}{l+1}\sum_{i=0}^{l}{|h}_{1440-i}^{*}-h_{1440-i}|$$

## 4. Results

For the estimated models, complexities (i.e., the value of m—higher m indicates more terms that are considered in the model, and hence the model becomes more complex), accuracies (i.e., the mean absolute difference between the last 20% of the predicted and the actual heartrates), and the model parameters are provided in Table 2. An indicative depiction of model convergence over iterations is shown in Figure 2, drawn for Participant 3. Similar parameter convergence was observed across all patients. An indicative model performance is depicted in Figure 3 against the actual heartrate of Participant 3 in the last 20% of the instances. Similar performance was observed in all participants. Presented in Table 3 is an indicative and preliminary clustering of patients that can be drawn from the current observations.

## 5. Discussion and Conclusions

A means for estimating underlying models of hearts was presented. This was performed using the first dataset collected from the pilot phase of the SmartHeart study. The SmartHeart study focused on testing a wearable, at-home, sensor-based digital self-monitoring system for people with heart failure. The estimated models were of an autoregressive architecture. The model estimation is computationally efficient, as we can estimate a model for the heartrate every minute. The models are estimated based on previous heartrate observations (no more than 1440 previous minutes). The models are estimated through the least norm solution. The estimated models predict the heartrate of the subsequent minute using at most five previous heartrates. After the first set of observations, the same analysis can be carried on to the future, in each minute, while maintaining 1440 as an observation window. The significance of the number 1440 comes as it covers a period of one day, i.e., 24 h × 60 min.

Using the first 1440 heartrate observations from each of our 23 participants, we could observe similarities and differences in the estimated models, as shown by the parameters in Table 2. Based on these parameters, we performed a preliminary clustering of participants based on the complexity of the best-fitting models that could be uncovered for each participant. This clustering that we have presented is indicative only. That means, we are only demonstrating the feasibility of capturing some diversity within the participants through the estimated models, but we are unable to comment on any clinical relevance or non-relevance of this diversity at this stage because we have not correlated these models with any significant cardiac events. Later, with more data and other observations such as symptoms and cardiac events, there could be so many other classification, clustering, risk assessment, and prediction tasks that could be performed [12]. As such, the clinical use of what we have proposed will be to serve as a tool that monitors participants over time and predicts any adverse events before they occur. Such predictions can trigger warnings to relevant stakeholders, such as clinicians, carers, and emergency services, thereby enabling participants to receive the right care at the right time to ensure preservation of good health and quality of life. Such predictions and warnings will be essential for high-quality hospital in the home (HITH) care models [16].

Even from the basic clustering presented in this paper, we could see some preliminary intra-class similarities and inter-class differences, such as the trends in signs and magnitudes of the model parameters, as indicated in Table 2. The *p*-values estimated for all parameters in Table 2 were near-zero, indicating the statistical significance of the estimated models. The model architecture can be made as complex or simple as desired, but maintaining simplicity, as we have done, helps in making the problem computationally and analytically tractable while being implementable for real-time modeling and analysis. Although more complex models might increase the agreement of the models with the heartrate data, they might be disadvantageous, as they can lean towards overfitting while making it more difficult to analyze and make sense of. Our approach of limiting the analysis to at most six parameters, we believe, is elegant in terms of reducing overfitting and enhancing interpretability. Given the computational simplicity of our approach, it can be implemented to estimate and keep track of model parameters of each person in real time. Estimating and keeping track of model parameters as such can have several advantages, as they can eventually serve as proxies that may be predictive of a person’s health, longevity, or impending health events.

There are several limitations in this paper. At this stage, we have only 23 participants, who have produced more than 1440 observations; therefore, the sample size in which we tested our approach was 23. We consider this sample size to be small to draw validated prediction models or statistically significant conclusions. Therefore, our intention at this stage is to not present validated prediction models of clinically relevant cardiac events. Our intention is to demonstrate the feasibility of using noninvasive wearable sensors, coupled with efficient computation, to capture the underlying models of the heart that could predict the heartrate within an error of no more than 2–7 beats per minute in most cases. Once this feasibility is established, approaches like this can be employed for longer-term studies to observe participants over a long period of time and identify statistically significant metrics that could predict clinically relevant cardiac events before they happen. We have also not correlated the heartrate with physical activity and heart failure symptoms of the participants observed, since we did not have adequate data, as our observation window was one day. Such a correlation will be insightful, as it is known that the heartrate has a direct correlation with, for instance, physical activity. Accounting for physical activity and symptoms may require more physiologically grounded nonlinear models. A challenge with that analysis is with how to collect the physical activity of a person to match the heartrate. Doing this would require an approach like actively encouraging study participants to wear a wearable device like a smartwatch throughout the day. We will explore this in future work.

In future work, we aim to take the parameters estimated for a person in real time and feed them to a classifier that might predict any risk of health events that might occur within the next 24 h or so. This will also give the opportunity to compare machine learning predictions with any existing baseline metrics such as mean heart rate, heart rate variability, or existing clinical risk scores. Doing this will require observing participants over a long period of time so that cardiac events and symptoms can be correlated with what is measured through sensors. Assuming unknown population size and varying effect sizes, a sample of about 300 to 400 participants will be required to be observed over a long period of time for a study like this. There is also an opportunity to incorporate certain nanoengineered biosensors [17]; however, any risk of side-effects must be monitored and considered. Implementation of such monitoring approaches can have immense implications in being able to prolong the lives of people and preserve the quality of life while reducing costly hospital admissions. The nature of the problem and our approach give us a rare opportunity to test and validate machine learning approaches in a healthcare context while performing thorough accuracy and bias analyses, not immediately, but over time.

Our approach is applicable to monitoring many chronic conditions, with heart failure studied in this paper a case study. An approach like ours can be implemented with more data collected over time for other health conditions and maybe different organs also. Such rich data collection and analysis can eventually help in creating a multidimensional and comprehensive model of a human, which could be considered a digital twin. Such a digital twin can be made use of in healthcare, as depicted in Figure 4.

Given the escalating costs of healthcare, the need to focus on prevention and tailored strategies for monitoring and management becomes increasingly important. We contend that this can only be achieved efficiently and effectively in a sustained approach by harnessing the full potential of IoT, coupled with advances in AI and machine learning, to develop digital twins to support hyper-personalization and precision.

## Figures and Tables

**Figure 1 sensors-26-00082-f001:**
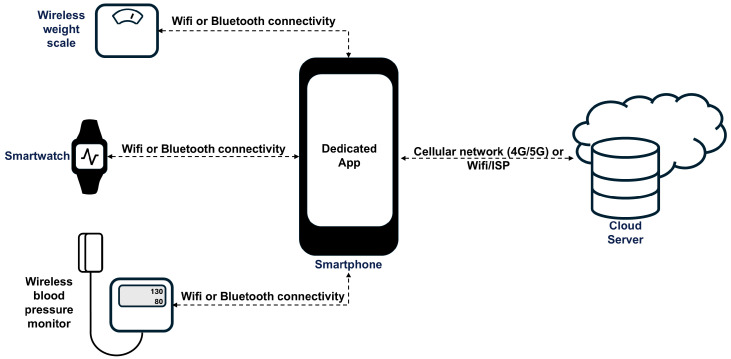
The sensor setup used.

**Figure 2 sensors-26-00082-f002:**
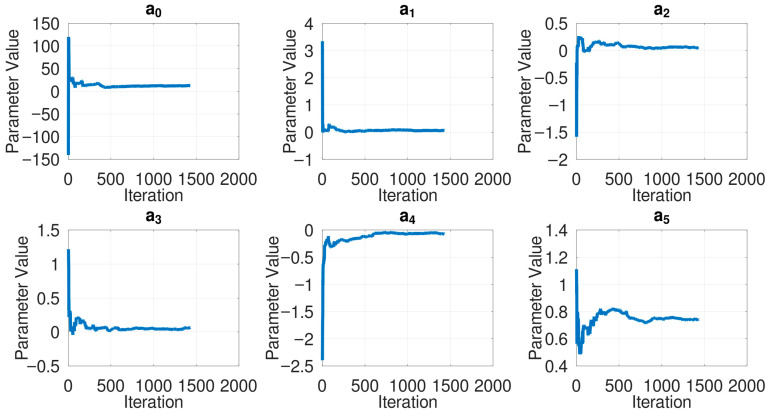
Parameter convergence for Participant 3—$a_{0}$ to $a_{5}$ denote model parameters; their variation over time is plotted as they are estimated every minute (Note: Certain values approaching zero indicate that they become small in magnitude, but they do not become exactly zero in value).

**Figure 3 sensors-26-00082-f003:**
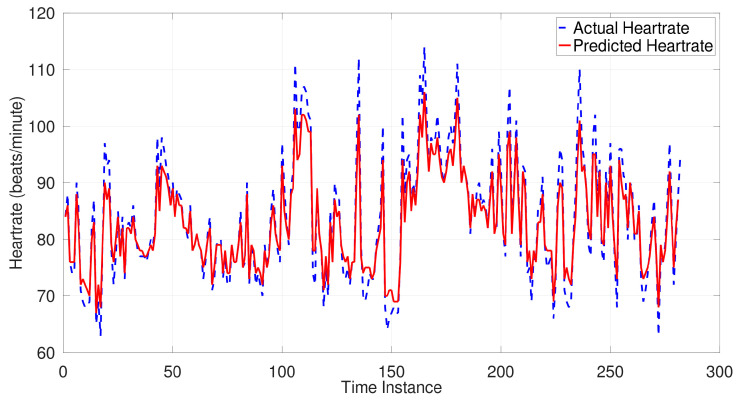
Indicative performance of a best-fitting model: Variation in the predicted heartrate alongside the actual heartrate for Participant 3.

**Figure 4 sensors-26-00082-f004:**
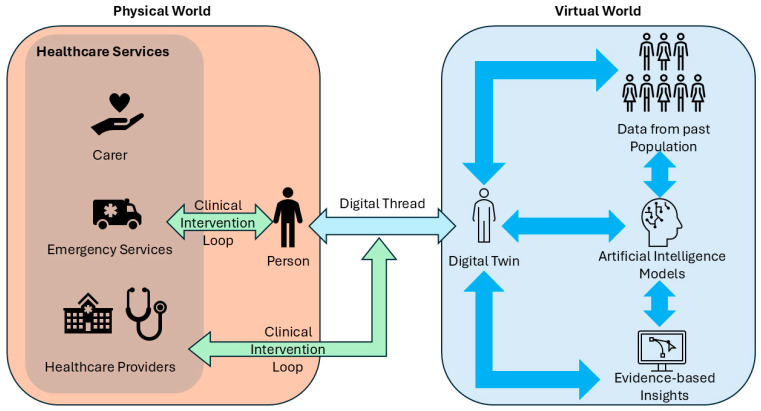
A healthcare-oriented digital twin paradigm to monitor people over time.

**Table 1 sensors-26-00082-t001:** The sensors used and the variables measured.

Sensor Type	Commercial Provider	Parameter(s) Measured
Wireless blood pressure monitor	Withings™ (Issy-les-Moulineaux, France)	Blood pressure
Smartwatch	Samsung Galaxy Watch5 Pro™ (Suwon, Republic of Korea)	Heartrate and physical activity
Wireless weight scale	Withings Body Smart Scale™	Body weight

**Table 2 sensors-26-00082-t002:** Details of the heart models that were estimated for the 23 participants (*p*-values observed for all parameters were near-zero, indicating statistical significance).

Participant ID	Complexity of Best Model (i.e., *m*)	Error of the Best Model (i.e., *Err*) in Beats per Minute	Parameter Values (Mean (±Standard Deviation))					
			a_0_ (±σ)	a_1_ (±σ)	a_2_ (±σ)	a_3_ (±σ)	a_4_ (±σ)	a_5_ (±σ)
Participant 1	5	6.84	5.97 (+/−0.348)	0.0543 (+/−0.00571)	0.0272 (+/−0.0164)	0.121 (+/−0.0191)	−0.0997 (+/−0.00965)	0.824 (+/−0.0145)
Participant 2	1	3.18	6.55 (+/−0.121)	0.916 (+/−0.00173)	0	0	0	0
Participant 3	5	6.32	12.1 (+/−0.269)	0.0588 (+/−0.00419)	0.0591 (+/−0.00556)	0.0453 (+/−0.00908)	−0.0606 (+/−0.0066)	0.744 (+/−0.00425)
Participant 4	5	4.49	7.9 (+/−0.471)	0.0492 (+/−0.00507)	−0.0058 (+/−0.0033)	0.119 (+/−0.00178)	−0.115 (+/−0.00209)	0.85 (+/−0.00445)
Participant 5	5	7.07	22.2 (+/−0.674)	0.0588 (+/−0.00496)	−0.0311 (+/−0.00622)	0.0853 (+/−0.00658)	−0.141 (+/−0.00743)	0.791 (+/−0.0115)
Participant 6	3	2.66	6.16 (+/−0.0851)	0.119 (+/−0.00703)	−0.0215 (+/−0.00754)	0.82 (+/−0.00353)	0	0
Participant 7	2	6.57	12.2 (+/−0.531)	−0.0176 (+/−0.0125)	0.863 (+/−0.0134)	0	0	0
Participant 8	3	5.67	17.2 (+/−0.391)	0.1 (+/−0.00898)	−0.00173 (+/−0.0139)	0.664 (+/−0.0086)	0	0
Participant 9	2	5.36	13.6 (+/−0.289)	0.0823 (+/−0.00776)	0.726 (+/−0.00541)	0	0	0
Participant 10	3	2.56	26.8 (+/−0.316)	0.144 (+/−0.00204)	−0.0494 (+/−0.00812)	0.574 (+/−0.00454)	0	0
Participant 11	5	7.12	39.7 (+/−10.1)	0.018 (+/−0.0218)	0.067 (+/−0.014)	0.116 (+/−0.0193)	−0.026 (+/−0.0246)	0.393 (+/−0.0279)
Participant 12	3	7.03	16.1 (+/−0.161)	0.0699 (+/−0.00777)	−0.0265 (+/−0.00434)	0.765 (+/−0.00351)	0	0
Participant 13	5	6.68	10.7 (+/−0.293)	0.0499 (+/−0.00586)	−0.00481 (+/−0.00629)	0.0387 (+/−0.00974)	0.0275 (+/−0.00379)	0.753 (+/−0.0104)
Participant 14	5	2.69	34.8 (+/−0.855)	0.0981 (+/−0.00334)	−0.00785 (+/−0.00571)	0.132 (+/−0.00421)	−0.00806 (+/−0.00563)	0.406 (+/−0.00631)
Participant 15	3	3.42	18.2 (+/−0.401)	0.0974 (+/−0.00916)	0.0106 (+/−0.00738)	0.657 (+/−0.0131)	0	0
Participant 16	5	6.55	15.3 (+/−0.779)	0.0208 (+/−0.00675)	−0.0334 (+/−0.00252)	0.094 (+/−0.00623)	0.202 (+/−0.00602)	0.526 (+/−0.00609)
Participant 17	5	3.98	15.1 (+/−0.314)	0.0729 (+/−0.00295)	−0.031 (+/−0.00323)	0.156 (+/−0.0052)	0.0346 (+/−0.00777)	0.577 (+/−0.00363)
Participant 18	2	4.09	21.9 (+/−0.198)	0.0283 (+/−0.00339)	0.699 (+/−0.00302)	0	0	0
Participant 19	4	7.22	10.6 (+/−0.211)	0.0413 (+/−0.0131)	0.101 (+/−0.016)	−0.0816 (+/−0.0131)	0.808 (+/−0.0128)	0
Participant 20	2	10.1	24.6 (+/−0.42)	0.155 (+/−0.00808)	0.542 (+/−0.00527)	0	0	0
Participant 21	1	8.44	17.2 (+/−0.197)	0.785 (+/−0.00222)	0	0	0	0
Participant 22	5	4.12	17 (+/−0.19)	0.0855 (+/−0.00223)	−0.0608 (+/−0.00498)	0.104 (+/−0.00288)	0.159 (+/−0.0123)	0.492 (+/−0.00516)
Participant 23	1	5.14	90.3 (+/−0.302)	−0.00243 (+/−0.00336)	0	0	0	0

**Table 3 sensors-26-00082-t003:** Indicative clustering based on best-fitting model complexity (i.e., value of m).

Cluster:	Cluster 1	Cluster 2	Cluster 3	Cluster 4	Cluster 5
Complexity of the best-fitting model (i.e., the value of m):	1	2	3	4	5
Number of Participants:	3	4	5	1	10
List of Participants:	Participant 2 Participant 21 Participant 23	Participant 7 Participant 9 Participant 18 Participant 20	Participant 6 Participant 8 Participant 10 Participant 12 Participant 15	Participant 19	Participant 1 Participant 3 Participant 4 Participant 5 Participant 11 Participant 13 Participant 14 Participant 16 Participant 17 Participant 22

## Data Availability

The data presented in this study are available on request from the corresponding authors due to the data governance requirements as stipulated in our ethics and governance protocols.

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
