# Peer review of "Time Series Models of the Human Heart in Patients with Heart Failure: Toward a Digital Twin Approach"

_sensors, 2025, doi:10.3390/s26010082_

Round 1
Reviewer 1 Report
Comments and Suggestions for Authors
This manuscript presents a novel, actual application of digital twin (DT) technology in the field of healthcare, specifically focusing on modeling heart rate time series data in patients affected by heart failure. Using data from the SmartHeart study, a computationally efficient autoregressive modeling approach to estimate heart rate dynamics is proposed, exploring its potential for real-time monitoring and personalized care.
The work has several strengths, including the area investigated, the sound and well-described methodology, the usage of real-world data that adds credibility and generalizability to the findings, and the overall structure of the paper, where also future directions and tips are highlighted.
Nevertheless, an improvement in model validation, including further metrics in this regard, is warmly welcomed, and also some practical suggestions for the use of the results in clinical practice. I also suggest pointing out the main study limitations, and carrying out a thorough revision of English and grammar to enhance correctness and readability.
Author Response
Thank you very much for the thorough review and valuable feedback. I have attempted to address the comments to the best of my ability. However, there are some comments that we cannot address at this time because we report from a pilot study. Once the pilot study is published, we can build on this work towards a completed study through which we could hopefully have all the answers. Please read through the attached pdf containing the responses to all the comments from all the reviewers.
Best regards,
Nalika.

Reviewer 2 Report
Comments and Suggestions for Authors
This manuscript presents a pilot study using wearable sensor data (heart rate, blood pressure, weight) from 23 heart failure patients to develop autoregressive (AR) time series models intended as "digital twins" for predicting heart rate dynamics. The authors estimate AR models of varying complexity (orders 1–5) using 1,440 minutes of heart rate data per participant and propose that these models could eventually forecast decompensation events.
1,With only 23 participants, the study lacks statistical power to support generalizable conclusions about heart failure heterogeneity. How do the authors justify that n = 23 is sufficient to capture clinically meaningful inter-patient variability in decompensation patterns, and what is the minimum sample size calculation (e.g., based on expected effect sizes in parameter variance) required to validate the claim that model complexity clustering reflects true physiological phenotypes rather than random noise?
2,The study uses the same 24-hour dataset for both model training and evaluation without temporal split-validation or out-of-sample testing. How did the authors verify that the estimated AR parameters remain stable over multiple days or weeks, and what is the temporal cross-validation strategy (e.g., forward-chaining) to ensure the models do not overfit to diurnal variations or transient artifacts rather than underlying pathophysiology?
3, The AR model assumes linear, stationary dynamics, yet heart rate in heart failure exhibits non-stationary, nonlinear behavior (e.g., circadian rhythms, autonomic dysfunction, ectopic beats, physical activity confounding). What evidence is provided that residuals are white noise and that the AR coefficients are physiologically interpretable (e.g., mapping to autonomic tone) rather than mere mathematical fits? Why were more physiologically grounded models (e.g., with exogenous inputs for activity or non-linear terms) excluded?
4,The manuscript does not correlate model parameters with any clinical ground truth—no decompensation events, symptom scores, BNP levels, hospitalizations, or ejection fraction changes are referenced. How can the authors claim these are "models of decompensation" without demonstrating any association between parameter drift, prediction error (Err), or model complexity and actual heart failure deterioration? What is the planned framework for prospective validation against clinical events?
5,The term "digital twin" implies a bidirectional, causal, and mechanistically integrated model that mirrors the physical system's state and responses to interventions. A unidirectional AR model derived from a single variable (heart rate) lacks causality, state awareness, and therapeutic feedback loops. Can the authors explicitly define their digital twin ontology and justify how this AR model meets the rigor of digital twin standards in healthcare (e.g., FDA's guidance on digital twins, or the seven core DT principles outlined by Iliuta et al., 2024)?
6,Wearable heart rate data are prone to artifacts (motion, poor contact) and missing values. The manuscript does not describe preprocessing, artifact removal, or how missing data were handled. What percentage of the 1,440 minutes were interpolated or excluded, what algorithms were used for artifact detection, and how does data completeness vary across participants? Could participants' adherence patterns (e.g., watch removal during sleep) bias the AR parameter estimates?
7,Even if models achieve low prediction error (Err ≈ 2–7 bpm), this error magnitude may be clinically irrelevant for preempting decompensation. What is the minimum clinically important difference (MCID) in heart rate prediction that would trigger a clinical action, and how does the proposed real-time implementation integrate into existing heart failure management workflows? Specifically, would clinicians be expected to act on parameter drift alone, and what false-alarm rate is acceptable in a home-monitoring context?
Author Response

(The authors gave the same response as above.)

Reviewer 3 Report
Comments and Suggestions for Authors
This study reports on time series models of the human heart in patients with heart failure using a digital twin approach. I would recommend this manuscript for publication in Sensors if the following points are adequately addressed.
- The study uses only 23 participants with 24 hours (1440 minutes) of heart rate data each. Can the authors justify how this limited sample size and short duration are sufficient to establish reliable digital twin models for heart failure patients?
- The manuscript presents autoregressive models but does not correlate the model parameters or predictions with actual clinical outcomes (e.g., heart failure decompensation events, hospitalizations, symptom severity). Can the authors provide evidence that these models have clinical relevance beyond mathematical fit?
- The authors acknowledge that heart rate has not been correlated with physical activity, which is a major confounding factor. How can the heart rate models be interpreted as indicators of heart failure status without accounting for physical activity?
- To strengthen the manuscript, please consider citing the following relevant studies: Bioengineering & Translational Medicine, 2025, e70002; Advanced Science, 2024, 11, 2400596; and Science Advances, 2024, 10, eadq6778.
- No comparison is made with simpler baseline models or existing heart failure monitoring approaches. Can the authors demonstrate that their autoregressive models outperform basic statistical measures (e.g., mean heart rate, heart rate variability) or existing clinical risk scores?
- Table 3 presents clustering based on model complexity, but no statistical analysis is provided. Are the differences between clusters statistically significant? What clinical or demographic characteristics distinguish patients in different clusters?
Author Response

(The authors gave the same response as above.)

Round 2
Reviewer 2 Report
Comments and Suggestions for Authors
accept